# *Lactobacillus fermentum* SMFM2017-NK4 Isolated from Kimchi Can Prevent Obesity by Inhibiting Fat Accumulation

**DOI:** 10.3390/foods10040772

**Published:** 2021-04-04

**Authors:** Doyeon Kim, Yukyung Choi, Sejeong Kim, Jimyeong Ha, Hyemin Oh, Yewon Lee, Yujin Kim, Yeongeun Seo, Eunyoung Park, Joohyun Kang, Yoonjeong Yoo, Soomin Lee, Heeyoung Lee, Yohan Yoon

**Affiliations:** 1Department of Food and Nutrition, Sookmyung Women’s University, Seoul 04310, Korea; rlaqhdus__@naver.com (D.K.); yw0322@naver.com (Y.L.); yujinkim77@gmail.com (Y.K.); sye0360@naver.com (Y.S.); scpey@naver.com (E.P.); diana_22@naver.com (J.K.); hellorunjing@gmail.com (Y.Y.); 2Risk Analysis Research Center, Sookmyung Women’s University, Seoul 04310, Korea; sjkim_11@naver.com (S.K.); hayan29@naver.com (J.H.); odry0731@naver.com (H.O.); slee0719@naver.com (S.L.); 3Food Standard Research Center, Korea Food Research Institute, Wanju 55365, Korea; hylee06@kfri.re.kr

**Keywords:** probiotics, lactic acid bacteria, *Lactobacillus fermentum*, kimchi, anti-obesity

## Abstract

This study evaluated the anti-obesity effects of lactic acid bacteria. Thirty-one lactic acid bacteria were examined in vitro for their ability to inhibit α-glucosidase activity, lipase activity, and 3T3-L1 cell differentiation. Four selected lactic acid bacteria were administered to obese C57BL/6J mice models for 8 weeks. The degree of improvement in obesity was determined by weight gain and serum biochemical analysis. The expression levels of genes (*Fas* and *Cpt-2*) related to obesity in the liver were analyzed by quantitative reverse transcription (qRT)-PCR. In addition, antioxidant protein levels (SOD-2, CAT, and GPx-1) in the liver were evaluated. The lactic acid bacteria-treated groups (PPGK1, LFNK3, LPNK2, and LFNK4) showed lower weight increase rate than the control group. The total cholesterol (T-chol), triglyceride (TG), alanine aminotransferase (ALT), and aspartate aminotransferase (AST) levels in the blood serum of the LFNK4 group were the lowest among other groups, compared to the control group. The expression levels of lipid metabolism-related genes (*Fas* and *Cpt-2*) in the liver of the LFNK4 group were lower in *Fas* and higher in *Cpt-2* than in the control group. The antioxidant protein expression levels (SOD-2, CAT, and GPx-1) in the liver tissue were also higher in the LFNK4 group. These results indicate that *L. fermentum* SMFM2017-NK4 has anti-obesity effects.

## 1. Introduction

Obesity is a medical condition in which excess fat accumulates in the body, which can adversely affect health [1]. It is closely related to an imbalance between energy intake and expenditure [2]. Obesity has recently been classified as a disease rather than a symptom [3]. Worldwide, more than 300 million people are considered to be overweight, and the current obesity rate in Korea has increased from 31.7% in 2007 to 34.1% in 2017 for adults over 19 years of age [4].

Most of the medicines used to treat obesity have side effects such as increased blood pressure, abdominal pain, anxiety, constipation, insomnia, and headaches [5,6]. Functional foods can substitute for medicines, and several studies are being conducted [7]. Probiotics have been developed as a functional food. They are live microorganisms that exert various beneficial effects on the host [8,9], and the genus *Lactobacillus* is a representative heterogeneous group of lactic acid bacteria (LAB), which are frequently used as probiotics [10]. Studies have reported that lactic acid bacteria regulate expression of genes involved in energy homeostasis to suppress obesity [11].

The lactic acid bacteria *Weissella cibaria* WIKIM28 and *Weissella onfusa* WIKIM51 isolated from kimchi at the Kimchi Research Institute showed anti-obesity effect by inhibiting the differentiation of adipocytes and lipid accumulation, and reducing the expression of adipocyte differentiation-related factors [12,13]. In addition, as fat is deposited in the liver due to obesity, alanine aminotransferase (ALT) and aspartate aminotransferase (AST) levels, which are predictors for liver inflammation, increase in blood serum [14]. Kwon et al. (2004) [15] reported that ALT and AST levels decreased after administration of lactic acid bacteria to obese mice, thereby alleviating fatty liver.

The purpose of this study was to develop probiotics with lactic acid bacteria that have anti-obesity effects using obese mouse models.

## 2. Materials and Methods

### 2.1. Preparation of Lactic Acid Bacteria

Thirty-one lactic acid bacteria strains were used (Table 1), which were isolated from kimchi in South Korea [16]. One hundred microliters of 20% glycerol stock of each strain was cultured in 10 mL Lactobacilli MRS (de Man, Rogosa and Sharpe) broth (Becton, Dickinson and Company, Franklin lakes, NJ, USA) and incubated anaerobically at 37 °C for 24 h, using an anaerobic pack (Anaerogen^TM^, Oxoid, Hampshire, UK). Then, 100 μL of the culture medium was transferred into fresh 10 mL Lactobacilli MRS broth (Becton, Dickinson and Company) and incubated at 37 °C for 24 h. The cultured medium was precipitated by centrifuge at 1912× *g* and 4 °C for 15 min, and the cell pellets were washed twice with phosphate-buffered saline (PBS; pH 7.4; KH_2_PO_4_ 0.2 g, Na_2_HPO_4_ 1.5 g, NaCl 8.0 g, KCl 0.2 g/distilled water 1 L). The final cell pellets were suspended in 10 mL of PBS to obtain 9–10 log CFU/mL.

### 2.2. Measurement of α-Glucosidase Inhibition Activity

The inhibitory effect of lactic acid bacteria on α-glucosidase activity was analyzed according to the method described by Lordan et al. (2013) [17]. Ten microliters of 1 U/mL α-glucosidase (Sigma-Aldrich, St. Louis, MO, USA) solution and 50 μL of 100 mM potassium phosphate buffer (pH 6.8) were mixed and 20 μL aliquots of the lactic acid bacteria strains were then added. The mixture was incubated at 37 °C for 15 min and 20 μL of 5 mM p-nitrophenyl α-D-glucopyranoside (Acros Organics, Waltham, MA, USA) was added and mixed well. After reacting at 37 °C for 20 min, 50 μL of 0.1 M Na_2_CO_3_ (Duksan Pure Chemicals Co., Ltd., Ansan, Gyeonggido, Korea) was added to stop the reaction. In a blank of the sample, lactic acid bacteria were added without α-glucosidase. Lactic acid bacteria were not added in a control sample (with α-glucosidase) and a blank of the control sample (without α-glucosidase). The reactants were placed into a 96-well plate (SPL Life Sciences, Pocheon, Gyeonggido, Korea) and absorbance was measured three times for each sample at a wavelength of 405 nm using an Epoch^TM^ microplate spectrophotometer (BioTek Instruments, Inc., Winooski, VT, USA).
Inhibition activity (%) = {1 − (Abs_S_ − Abs_SB_)/(Abs_C_ − Abs_CB_)} × 100(1)

Abs_S_: absorbance of a sampleAbs_SB_: absorbance of a blank of the sampleAbs_C_: absorbance of a control sampleAbs_CB_: absorbance of a blank of the control sample

### 2.3. Measurement of Pancreatic Lipase Inhibition Activity

The inhibitory effect of lactic acid bacteria on pancreatic lipase activity was analyzed according to the method described by Ana and Thomas (2019) [18]. Five hundred microliters of Tris-HCl buffer (Tech&Innovation, Chuncheon, Gangwondo, Korea), adjusted to pH 8.5, and 12.5 μL of 5 mM 4-nitrophenyl octanoate (NPC; Alfa Aesar, Haverhill, MA, USA) were added and mixed with 125 μL of the inoculum. Then, 12.5 μL of pancreatic lipase solution (5 mg/mL) (Sigma-Aldrich) was added and mixed for 2 min without forming bubbles. The mixed solution was incubated at 37 °C for 30 min, and 150 μL was placed into a 96-well plate (SPL Life Sciences). In a blank of the sample, lactic acid bacteria were added with dimethyl sulfoxide (DMSO, Sigma-Aldrich) instead of NPC. Lactic acid bacteria were not added in a control sample (with NPC) and a blank of the control sample (with DMSO instead of NPC). The absorbance was measured three times for each sample at a wavelength of 412 nm using an Epoch^TM^ Microplate Spectrophotometer (BioTek Instruments, Inc.), and the inhibition activity was calculated as follows:Inhibition activity (%) = 100 − {(Abs_S_ − Abs_SB_)/(Abs_RC_ − Abs_B_) × 100}(2)

Abs_S_: absorbance of a sampleAbs_SB_: absorbance of a blank of the sampleAbs_RC_: absorbance of a control sampleAbs_B_: absorbance of a blank of the control sample

### 2.4. Measurement of Inhibition for 3T3-L1 Adipocyte Differentiation

3T3-L1 preadipocyte cells were purchased from the Korean Cell Line Bank (KCLB; Seoul, Korea). The cells were cultured in Dulbecco’s modified Eagle’s medium (DMEM; Hyclone, Logan, UT, USA) supplemented with 10% newborn calf serum (NBCS; Gibco, Thermo Fisher Scientific, Waltham, MA, USA) and 1% penicillin–streptomycin solution (PS; Oxoid, Basingstoke, Hampshire, UK) in a 72T flask (Corning Inc., New York, NY, USA) and incubated at 37 °C and 5% CO_2_ for 2 days. Cultured cells were then sub-cultured in fresh medium for another 2 days and washed with Dulbecco’s phosphate-buffered saline (DPBS; Welgene, Gyeongsan, Gyeongsangbukdo, Korea). The cells were then detached with 0.05% trypsin-0.02% EDTA (Gibco). Then, the cells were centrifuged at 1000 rpm and 25°C for 5 min and mixed with the medium to be used in the experiment. 3T3-L1 cells were dispensed into 12-well plates (SPL Life Sciences) at a concentration of 2 × 10^4^ cells/mL, and cultured at 37 °C with 5% CO_2_. 3T3-L1 cells were cultured in MDI medium (DMEM; Hyclone), 10% fetal bovine serum (FBS; Gibco), 1% penicillin–streptomycin solution (PS; Oxoid), and 5 μg/mL insulin for 2 days to differentiate into adipocytes. The medium was then replaced with fresh growth media (DMEM, 10% FBS, and 1% PS) containing the lactic acid bacteria strains and refed every 2 days for four times to complete differentiation. The differentiated cell cultures were washed three times with DPBS (Welgene) and treated with 10% formalin for 1 h to fix the cells. The differentiated cells with spherical shape were stained with oil red O (Sigma-Aldrich) solution for 30 min in the dark and washed twice with distilled water. The cells were dissolved in isopropyl alcohol (Duksan Pure Chemicals Co., Ltd.) and absorbance was measured three times at a wavelength of 520 nm using an Epoch^TM^ Microplate Spectrophotometer (BioTek Instruments, Inc.). The inhibitory activity was calculated using the following equation:(3)$$\mathrm{Inhibition}\;\mathrm{activity}\;(\%)\;=\;\frac{\mathrm{Control}\;O.D.-\mathrm{Sample}\;O.D.}{\mathrm{Control}\;O.D.}\;\times \;100$$

The control optical density (O.D.) is the absorbance of the test solution with PBS instead of the bacterial solution, and sample O.D. shows the absorbance of the test solution with bacterial solution.

### 2.5. Development of Obese Mouse Model and Administration of Lactic Acid Bacteria

Of 31 lactic acid bacteria, 10 strains were ranked for high α-glucosidase inhibition activity and pancreatic lipase inhibition activity, and 4 strains commonly ranked for both enzyme inhibition activities were selected. The strains were identified as *Pediococcus pentosaceus* SMFM2017-GK1, *Lactobacillus fermentum* SMFM2017-NK3, *Lactobacillus plantarum* SMFM2017-NK2, and *Lactobacillus fermentum* SMFM2017-NK4. For animal experiments, 48 male SPF C57BL/6J mice 4 weeks old were purchased from Raonbio Inc. (Yongin, Gyeonggido, Korea). The mice were adapted for a week at 18–26 °C and 40–60% humidity. Then, the mice were randomly divided into 6 groups of 8 mice each (normal (normal mice treated with PBS), control (obesity-induced mice treated with PBS), PPGK1 (obesity-induced mice treated with *P. pentosaceus* SMFM2017-GK1), LFNK3 (obesity-induced mice treated with *L. fermentum* SMFM2017-NK3), LPNK2 (obesity-induced mice treated with *L. plantarum* SMFM2017-NK2), and LFNK4 (obesity-induced mice treated with *L. fermentum* SMFM2017-NK4) groups). Obesity in mice was induced by supplying a 60% high-fat diet (Research Diets, Inc., New Brunswick, NJ, USA) for 6 weeks to the control, PPGK1, LFNK3, LPNK2, and LFNK4 groups. The normal group was fed a normal diet (Research Diets, Inc.). The body weight of the mice was measured every 3 days. Two hundred microliters of lactic acid bacteria were administered orally to the obese mice in PPGK1, LFNK3, LPNK2, and LFNK4 groups, and 200 μL PBS was administered to the control and normal groups. The weight gain was calculated by comparing the difference between the weight after induction of obesity at 8 weeks and the weight after oral administration at 14 weeks. After 8 weeks of treatment, all the mice were sacrificed using CO_2_ gas. The animal experiments were performed after approval by the Institutional Animal Care and Use Committee of Sookmyung Women’s University (approval number: SMWU-IACUC-1912-027).

### 2.6. Analysis of Serum Cholesterol and Lipid

Blood was collected from the posterior aorta of the mice after sacrifice and placed in a serum separation tube (Microtainer; Becton, Dickinson and Company) for 30 min. Serum was obtained by centrifugation of the blood at 4 °C and 5000 rpm for 10 min and stored at −70 °C. Blood serum was analyzed using a Mindray BS-220 biochemistry analyzer (Mindray Medical International Ltd., Shenzhen, China).

### 2.7. Pathological Analysis of Adipose and Liver Tissue

The epididymal fat and the liver were fixed with 10% formalin (Hisko, Gunpo, Gyeonggido, Korea), and a paraffin block was prepared to cut 3–4 μM sections. The adipose tissues of the epididymal fat and the liver tissues were observed using a microscope (Leica, Wetzlar, Germany) after hematoxylin and eosin (H&E) staining, and the size and area of adipocytes and steatosis of hepatocytes were analyzed using the ImageJ program (National Institutes of Health, Bethesda, MD, USA).

### 2.8. Transcriptome Analysis of the Liver Tissue

To investigate the mRNA expression in the liver tissues, liver mRNA was extracted using an RNeasy mini kit (Qiagen, Hilden, Germany) according to the manufacturer’s instructions. Complementary DNA (cDNA) was synthesized using the extracted mRNA, according to the manufacturer’s instructions, using a QuantiTect Reverse Transcription kit (Qiagen). cDNA was then used to evaluate the gene expression by quantitative reverse transcription PCR (qRT-PCR) and a Rotor-Gene SYBR^®^ Green PCR kit (Qiagen), and qRT-PCR was performed using a Rotor-Gene Q (Qiagen). In brief, 25 μL reaction mixture was used comprising 1 μL template cDNA, 12.5 μL 2 × rotor-gene SYBR^®^ green PCR master mix, 6.5 μL RNase-free water, 2.5 μL forward primer (10 pmol), and 2.5 μL reverse primer (10 pmol). The amplification conditions were as follows: 5 min preheating at 95 °C, followed by 35 cycles of 5 s denaturation at 95 °C, and 10 s annealing at 60 °C. Gene expression levels of *β**-actin*, *Fas*, and *Cpt-2* were analyzed using specific primers for each gene (Table 2), and the relative gene expression was calculated using the 2^−ΔΔ*Ct*^ method [19].

### 2.9. Immunoblotting

To analyze protein expression in the liver, protein was extracted from 30 to 50 mg of liver using a PRO-PREP^TM^ protein extraction solution (iNtRON Biotechnology Inc., Seongnam, Gyeonggido, Korea) according to the manufacturer’s instructions. The concentration of the extracted protein was quantified using a DC^TM^ Protein Assay (Bio-Rad Laboratories, Inc., Hercules, CA, USA). Protein expression was quantified by Western blotting as follows: 20 μg of protein was separated by 12% sodium dodecyl sulfate (SDS)-PAGE at 200 V for 30 min. The proteins were then transferred to a polyvinylidene difluoride (PVDF) membrane (GE Healthcare Life Sciences, Marlborough, MA, USA) at 60 V for 2.5 h. Membranes were incubated with 5% skim milk (Sigma-Aldrich) at room temperature (20 °C) for 1 h. The primary antibodies (mouse-anti-SOD-2 (sc-130345, 1:100), mouse-anti-CAT (sc-271803, 1:1000), and GPx-1 (sc-133160, 1:500)) (Santa Cruz Biotechnology Inc., Santa Cruz, TX, USA) were used for immunoblotting, and mouse-anti-*β-actin* (sc-81178, 1:1000, Santa Cruz Biotechnology Inc.) was used as a reference protein. Anti-mouse IgG HRP (sc-2005, 1:5000, Santa Cruz Biotechnologies, Inc., Dallas, TX, USA) was used as the secondary antibody. The immunoreactive bands were visualized with an LAS-3000 Imager (Fujifilm, Tokyo, Japan), and the intensity of the bands was quantified using GelQuant software v.2.7. (DNR Imaging Systems Ltd., Jerusalem, Israel).

### 2.10. Statistical Analysis

SAS version 9.4 (SAS Institute Inc., Cary, NC, USA) was used for the statistical analysis. Significant differences were determined between the two groups (normal and lactic acid bacteria-treated groups) using a general linear model (GLM) with a pairwise *t*-test at *α* = 0.05. The least-squares (LS) means were compared to analyze the differences.

## 3. Results/Discussion

### 3.1. Inhibition of α-Glucosidase Activity by Lactic Acid Bacteria

α-Glucosidase is an enzyme that catalyzes the last step in the digestion of carbohydrates, converting them to glucose [23,24]. α-Glucosidase inhibitors in the small intestine inhibit the process of disaccharide-to-monosaccharide degradation [25]. The activity of α-glucosidase inhibitors delays digestion and absorption of sugars in the intestine [26]. To select lactic acid bacteria that have anti-obesity effects, we examined whether they inhibit α-glucosidase activity. Table 3 shows the results of the inhibition of α-glucosidase activity by the 31 lactic acid bacteria strains. Thirty-one lactic acid bacteria strains showed 6.58–41.45% inhibitory activity against α-glucosidase. The top 10 strains with the highest α-glucosidase inhibitory activity were LAB 66 (41.45%), 29 (40.64%), 50 (39.13%), 70 (39.09%), 75 (37.22%), 45 (37.04%), 44 (34.64%), 73 (33.11%), 20 (30.68%), and 98-1-3 (30.16%). Among the 31 lactic acid bacteria, strain 66 showed the highest (41.45%) α-glucosidase inhibitory activity. EI-shiekh et al. (2019) [27] showed that the activities of various plant extracts including *Solenostemma argel* were 37.29%–89.08%. Kim and Lim (2020) [28] reported that α-glucosidase inhibitory activity of the milk fermented with *L. plantarum* KI134 strain was 2.74%, which was the highest among other 255 strains. Compared to these activities, the α-glucosidase inhibitory activity in our study presented above can be considered similar or high. Thus, it considered that the lactic acid bacteria strains may have an effect on obesity. Hence, further analysis was performed.

### 3.2. Inhibition of Pancreatic Lipase Activity by Lactic Acid Bacteria

Pancreatic lipase is an enzyme that hydrolyzes fat triacylglycerol into 2-monoacylglycerols and fatty acids [29]. Orlistat, a representative inhibitor of pancreatic lipase, has excellent efficacy and inhibits approximately 30% hydrolysis of the ingested fat; however, orlistat has side effects such as gastrointestinal disorders, irritability, bile secretion disorders, and inhibition of absorption of fat-soluble vitamins [30]. Recently, studies have been conducted to develop inhibitors derived from natural products without side effects [31]. Based on these findings, we examined the potential of 31 lactic acid bacteria to inhibit obesity by investigating their ability to inhibit pancreatic lipase activity. The results are presented in Table 4. The 31 lactic acid bacteria strains showed 10.58%–18.92% inhibitory activity against pancreatic lipase, and the top 10 lactic acid bacteria strains with the highest pancreatic lipase inhibitory activity were LAB 20 (18.92%), 11 (18.80%), 26 (18.10%), 30 (15.22%), 29 (14.43%), 73 (13.74%), 72 (10.96%), 50 (10.95%), 38-Y (10.39%), and 31 (10.15%). In the studies by Kang et al. (2018) [32], the lipase inhibitory activity of *Lactobacillus fermentum* MG4231 was 12.24%, similar to the results of this study, and it has been shown to have a good inhibitory effect on adipogenic differentiation. The results of the inhibition of α-glucosidase activity and pancreatic lipase activity by the lactic acid bacteria showed that LAB 20, 29, 50, and 73 strains had excellent inhibitory activities.

### 3.3. Inhibition of 3T3-L1 Cell Differentiation by Lactic Acid Bacteria

Triglycerides are accumulated during differentiation of 3T3-L1 pre-adipocyte in cells by various transcription factors such as peroxisome proliferator-activated receptor γ (PPARγ), CCAAT-enhancer binding protein α (C/EBPα), and hormones [33]. Several studies analyzed the effect of obesity prevention through the inhibition of 3T3-L1 cell differentiation. Kim et al. (2010) [34] evaluated the effect of soy milk fermented with *Bacillus subtilis* KC-3 on the inhibition of 3T3-L1 cell differentiation and found that the fermented soy milk reduced 19% of fat accumulation, and Samjeonghwan fermented with lactic acid bacteria reduced the number of adipocytes [35]. In a study by Kim and Lim (2017) [36], *L. plantarum* K6 was selected as a lactic acid bacterium that has an anti-obesity effect; it inhibited differentiation of 3T3-L1 by 27.4%. These are similar to the highest inhibitory value of 21.44% (strain LAB 73) obtained in our study (Table 5). The strains LAB 29, 20, 50, and 73 were identified and named as *L. fermentum* SMFM2017-NK3, *P. pentosaceus* SMFM2017-GK1, *L. plantarum* SMFM2017-NK2, and *L. fermentum* SMFM2017-NK4, respectively (Table 5).

### 3.4. Weight Gain of the Obese Mouse Model

Obesity was induced in mice by a 60% high-fat diet for 6 weeks, and the weight of the control group (38.32 g) was significantly higher than that of the normal group (25.81 g) (*p* < 0.05). After administration of lactic acid bacteria in each group (*P. pentosaceus* SMFM2017-GK1 (PPGK1), *L. fermentum* SMFM2017-NK3 (LFNK3), *L. plantarum* SMFM2017-NK2 (LPNK2), and *L. fermentum* SMFM2017-NK4 (LFNK4)), the weight gain, which is the difference between the maximum body weight after induction of obesity and the minimum body weight after administration of lactic acid bacteria, in all four groups (PPGK1 (9.51 g), LFNK3 (9.01 g), LPNK2 (9.35 g), and LFNK4 (9.28 g)) were similar (Figure 1). These weight gains were lower by 2.0 g (PPGK1), 2.5 g (LFNK3), 2.1 g (LPNK2), and 2.2 g (LFNK4), respectively, compared to the control group (*p* < 0.05). In a study by Park et al. (2018) [37], the obese C57BL/6J mice which were treated with lactic acid bacteria (combination of *Lactobacillus casei* WK3, *Enterococcus faecium* WK5, *Bifidobacterium longum* WK9, and *L. plantarum* K-1) for 6 weeks had lower weight gain of 0.4 g, compared to the control group. In addition, fat accumulation in epididymal adipose tissue and symptoms of fatty liver were found to be alleviated [37]. Thus, lower weight gains of 2.0–2.5 g would be effective against obesity, and lactic acid bacteria (*P. pentosaceus* SMFM2017-GK1 (PPGK1), *L. fermentum* SMFM2017-NK3 (LFNK3), *L. plantarum* SMFM2017-NK2 (LPNK2), and *L. fermentum* SMFM2017-NK4 (LFNK4)) might reduce fat accumulation.

Data are presented as mean and standard error. Normal: normal diet + phosphate-buffered saline (PBS); Control: 60% high fat diet (HFD) + PBS; PPGK1: HFD + *P. pentosaceus* SMFM2017-GK1; LFNK3: HFD + *L. fermentum* SMFM2017-NK3; LPNK2: HFD + *L. plantarum* SMFM2017-NK2; LFNK4: HFD + *L. fermentum* SMFM2017-NK4.

### 3.5. Indicators of Lipid and Inflammation-Related Markers in Blood Serum

After treatment of the obese mice with lactic acid bacteria for 8 weeks, the concentrations of serum total cholesterol (T-chol) were 116.6, 264.8, 227.0, 258.8 and 204.0 mg/dL in the normal, control, PPGK1, LPNK2, and LFNK4 groups, respectively (Figure 2a). The concentrations of triglycerides (TG) were 65.6 and 115.0 mg/dL in normal and control groups, respectively. After treatment with lactic acid bacteria, the concentrations of TG in the PPGK1 (87.2 mg/dL), LFNK3 (81.0 mg/dL), LPNK2 (79.7 mg/dL) and LFNK4 (72.2 mg/dL) groups were lower (*p* < 0.05) than that in the control group (115.0 mg/dL), and the level in the LFNK4 group was the lowest (*p* < 0.05) (Figure 2b). HDL-cholesterol (107.0 mg/dL) and LDL-cholesterol (6.8 mg/dL) levels in the normal group were lower (*p* < 0.05) than those of the control group (HDL-cholesterol: 190.3 mg/dL, LDL-cholesterol: 29.1 mg/dL). After treatment of lactic acid bacteria, HDL-cholesterol levels were 181.9, 203.7, 191.7 and 167.2 mg/dL in PPGK1, LFNK3, LPNK2 and LFNK4 (Figure 2c). LDL-cholesterol levels were 24.4, 28.6, 25.4 and 21.8 mg/dL in PPGK1, LFNK3, LPNK2 and LFNK4 groups, and the level in the LFNK4 group was the lowest among the groups administered lactic acid bacteria (Figure 2d). For ALT and AST, the levels were 41.2 U/L and 73.1 U/L in the normal group, which were lower (*p* < 0.05) than those in the control group (ALT: 171.0 U/L, AST: 161.5 U/L). The ALT levels were 147.9, 148.4, 144.9 and 102.3 U/L, and AST levels were 149.5, 154.2 U/L, 153.0 and 118.3 U/L in PPGK1, LFNK3, LPNK2 and LFNK4 groups, respectively, after lactic acid bacteria treatment (Figure 2d,e). These levels in the LFNK4 group were the lowest among the groups administered lactic acid bacteria. Based on these results, the *L. fermentum* SMFM2017-NK4 strain, which was administered to the LFNK4 group, is the most effective in reducing blood serum lipids, especially TG and alleviating inflammation-related markers among the four lactic acid bacteria.

### 3.6. Pathological Features of Adipose and Liver Tissue

In general, the size of epididymal adipose tissue fat cells increases significantly when obesity is induced by the intake of a high-fat diet [38]. Ryu et al. (2011) [39] reported that the high-fat diet group increased the size of adipocytes compared to the normal group, and the group fed with white beech mushroom (*Hypsizigus marmoreus*) powder decreased the size of adipocytes, thereby inhibiting fat accumulation in adipose tissue. In our study, the adipocytes of the mice in the control group, which were administered a high-fat diet, were larger and irregularly shaped, compared to those of the normal group in the same ×40 magnification (Figure 3). In the groups administered lactic acid bacteria, the size and shape of adipocytes were similar to those in the normal group (Figure 3). The average area of adipocytes increased by 189.7% in the control group compared to the normal group (*p* < 0.05), and it decreased in the group administered with lactic acid bacteria, compared to the control group. After treatment with lactic acid bacteria, the average area of adipocytes in the LFNK4 group (135.7%) was significantly decreased (*p* < 0.05) compared to the control group, followed by LFNK3 (137.1%), PPGK1 (162.3%) and LPNK2 (217.4%) groups (Figure 4). The average area of adipocytes in the LFNK4 group decreased the most by 154.0% compared to that of the control group. In a study by Song et al. (2016) [40], the average area of adipocytes increased significantly by 83.2% in the control group compared to the normal group and treatment with lactic acid bacteria resulted in a 29.0% decrease in the average area of adipocytes, compared to the control group [40]. Thus, it was confirmed that the *L. fermentum* SMFM2017-NK4 (LFNK4) was effective in decreasing fat synthesis and the size of fat cells.

In the liver tissue of obese mice, severe steatosis was observed according to the International Harmonization of Nomenclature and Diagnostic Criteria (INHAND) [41]. In a study by Yu et al. (2019) [42], the livers of mice fed a normal diet were normal, but fat particles were observed on the surface of the liver of mice fed a 60% high-fat diet for 14 weeks. In a study by Park et al. (2015) [43], fat deposition in the liver tissue was reduced in the group administered with lactic acid bacteria-fermented Dioscoreae Rhizoma powder, compared to the control group. In a study by Lee et al. (2017) [44], a large amount of fat was accumulated in the liver in the control group that was fed a high-fat diet, and liver fat in the group administered ethanol extract of *Aster yomena* was reduced. In our study, severe steatosis was observed in the control group, which had fat accumulation in the liver. After administration of the lactic acid bacteria, fewer lesions were observed in the LFNK3 and LFNK4 groups; however, many steatosis lesions were observed in the PPGK1 and LPNK2 groups (Figure 5) (Table 6). The degree of steatosis in the liver was severe in the control, PPGK1, and LPNK2 groups, and it was moderate in the LFNK3 and LFNK4 groups (Table 6). Our results showed that administration of lactic acid bacteria, especially *L. fermentum* SMFM2017-NK3 (LFNK3) and *L. fermentum* SMFM2017-NK4 (LFNK4), reduced fat accumulation in the liver.

### 3.7. Transcriptome in the Liver Tissue

*Fas* is an enzyme involved in the synthesis of fatty acids from acetyl-CoA and malonyl-CoA [45]. Fatty acid synthesis decreases if the expression levels of *Fas* are suppressed, which reduces accumulation [46]. In Ikarashi et al. (2011) [47], the level of *Fas* gene expression in the liver decreased in the group administered with acacia polyphenol compared to the high-fat diet group. In this study, the expression levels of *Fas* were significantly lower (*p* < 0.05) in the LFNK4 group than in the other lactic acid bacteria-treated groups, compared to the control group (Figure 6a). In detail, the *Fas* expression level in the control group increased by 1.9 times, compared to that of the normal group; however, the level in the LFNK4 group increased by 0.6 times, compared to the normal group (Figure 6a). In the case of a study by Kim et al. (2019) [48], the expression level of *Fas* gene in the liver increased by 1.7 times in the high-fat diet group, compared to that of the normal group. The level decreased by 1.3 times in the group administered with the mixture of *Atractylodes macrocephala* and *Amomum villosum* extracts, compared to the normal group. Thus, it is considered that *L. fermentum* SMFM2017-NK4 (LFNK4) was more effective on lowering *Fas* gene expression than the mixture of Kim et al. (2019) [48]. *Cpt-2* is required for the synthesis of carnitine palmitoyl transferase 2, which is an essential enzyme for fatty acid oxidation [49]. Fatty acids must move from cells to mitochondria to be subjected to degradation by β-oxidation, and *Cpt-2* is an enzyme that catalyzes this process [50]. Therefore, the oxidation rate of fatty acids increases when the expression of *Cpt-2* increases and body fat decreases [51]. In our study, the expression levels of *Cpt-2* appeared the highest in the LFNK4 group among the lactic acid bacteria-treated groups, compared to the control group, although there was no statistically significant difference (Figure 6b). Therefore, *L. fermentum* SMFM2017-NK4 (LFNK4) was effective on regulating gene expressions in the liver, especially *Fas* gene.

### 3.8. Protein Expression Related to Antioxidant Enzymes in the Liver Tissue

Based on the results of the in vivo experiments presented above, the LFNK4 bacteria were the most effective in improving obesity among the other lactic acid bacteria. Thus, the protein expression levels in the liver tissue of the mice of the various groups were further analyzed. There are many studies on antioxidant enzymes such as superoxide dismutase (SOD), catalase (CAT), and glutathione peroxidase (GPx). Obesity affects the activity of these enzymes resulting in abnormalities in the antioxidant defense mechanism and an increase in free radicals in the liver. For this reason, studies on the antioxidant system in the liver of obese animal models are being conducted. The expression levels of antioxidant-related proteins (SOD-2, CAT, and GPx-1) in the liver tissues were analyzed (Figure 7). The expression levels of SOD-2 and CAT appeared higher in the LFNK4 group than in the control group, although there was no statistically significant difference (Figure 7a,b), and the expression levels of GPx-1 were significantly increased (*p* < 0.05) in the LFNK4 group compared to the control group (Figure 7c). According to a study by Kim et al. (2019) [52], the expression levels of the antioxidant proteins SOD, CAT, and GPx were increased after treatment with Banggihwanggitang-hap-yeonggyechulgam-tang, a medicinal herb used to treat obesity, in an obese mouse model, compared to the group that did not receive treatment. Our study also showed that the expression levels of the antioxidant proteins, especially GPx-1, increased in the LFNK4 group compared to that in the control group. Therefore, *L. fermentum* SMFM2017-NK4 (LFNK4) was found to be effective at reducing obesity-induced oxidative stress in the liver.

As described in Figure 8, *L. fermentum* SMFM2017-NK4 downregulated the expression of the *Fas* gene, which resulted in decreased fat synthesis in the liver. Furthermore, it upregulated the protein expression of GPx-1, increasing the antioxidant capacity that was reduced by obesity. As a result, it affected lowering the concentration of TG in the blood serum, adipocyte surface area, and steatosis in the liver; thus, the weight gain decreased.

## 4. Conclusions

In conclusion, *L. fermentum* SMFM2017-NK4, which had inhibition ability for α-glucosidase and pancreatic lipase activity, and 3T3-L1 pre-adipocyte cell differentiation, decreased TG in the blood serum, the size of adipocytes, and the steatosis of the liver. It was affected by downregulating expression of the *Fas* and upregulating GPx-1. Thus, weight gain was eventually decreased. Therefore, *L. fermentum* SMFM2017-NK4 can alleviate obesity.

## Figures and Tables

**Figure 1 foods-10-00772-f001:**
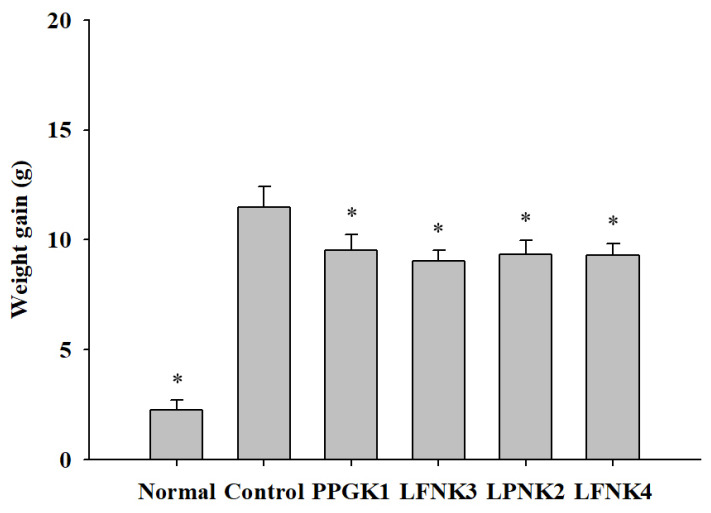
Weight gain (g) in mice after treatment of lactic acid bacteria for 8 weeks. * Statistically significant, compared to the control group by pairwise *t*-test (*p* < 0.05).

**Figure 2 foods-10-00772-f002:**
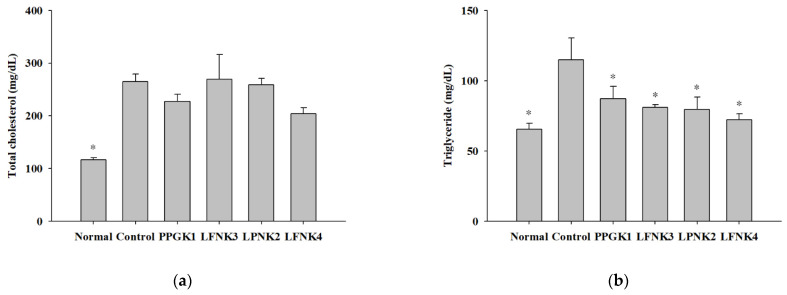
Serum level of total cholesterol (T-chol) (**a**), triglyceride (TG) (**b**), HDL-cholesterol (**c**), LDL-cholesterol (**d**), alanine aminotransferase (ALT) (**e**), and aspartate aminotransferase (AST) (**f**) in serum after the treatment of lactic acid bacteria for 8 weeks. Data are presented as mean and standard error. Normal: normal diet + phosphate-buffered saline (PBS); Control: 60% high-fat diet (HFD) + PBS; PPGK1: HFD + *P. pentosaceus* SMFM2017-GK1; LFNK3: HFD + *L. fermentum* SMFM2017-NK3; LPNK2: HFD + *L. plantarum* SMFM2017-NK2; LFNK4: HFD + *L. fermentum* SMFM2017-NK4. * Statistically significant, compared to the control group by pairwise *t*-test (*p* < 0.05).

**Figure 3 foods-10-00772-f003:**
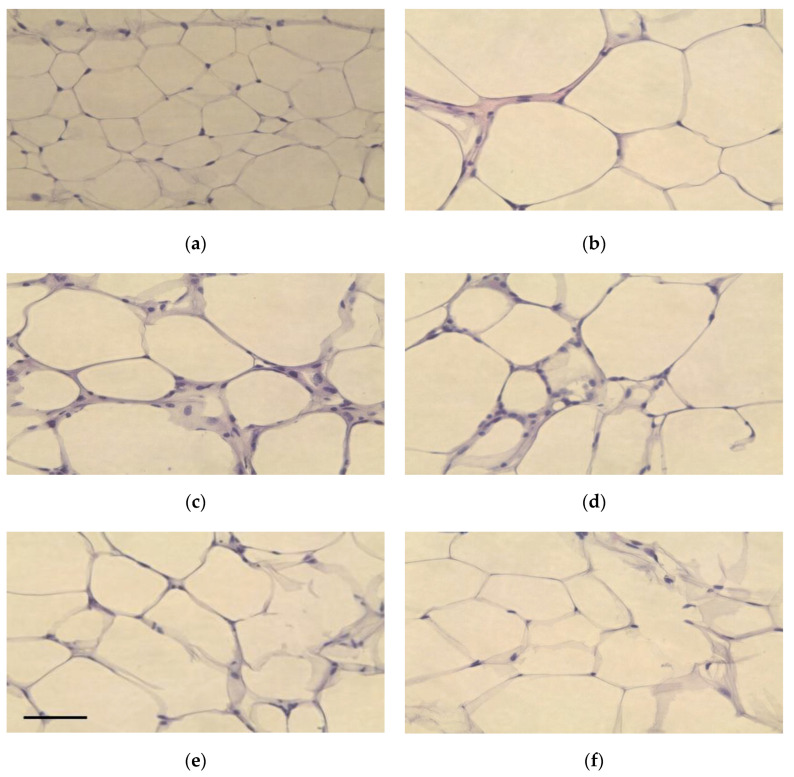
Representative microscopic images (×40) on epididymal white adipose tissues from mice groups. Scale bar, 50 μm. (**a**) Normal: normal diet + phosphate-buffered saline (PBS), (**b**) Control: 60% high-fat diet (HFD) + PBS, (**c**) PPGK1: HFD + *P. pentosaceus* SMFM2017-GK1, (**d**) LFNK3: HFD + *L. fermentum* SMFM2017-NK3, (**e**) LPNK2: HFD + *L. plantarum* SMFM2017-NK2, (**f**) LFNK4: HFD + *L. fermentum* SMFM2017-NK4.

**Figure 4 foods-10-00772-f004:**
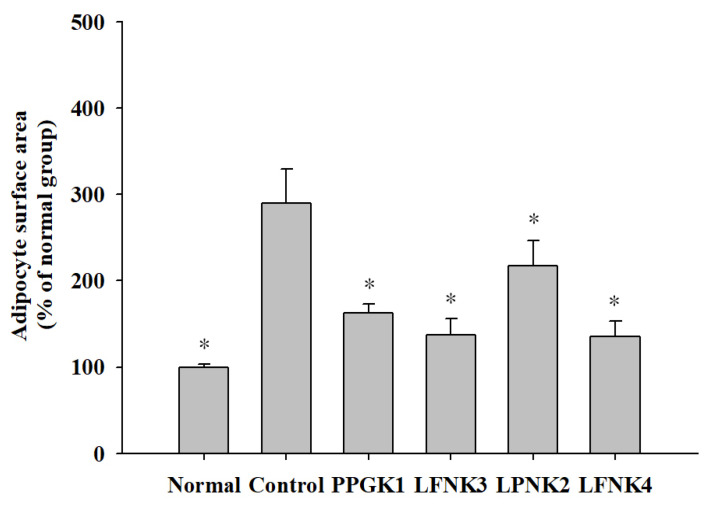
Epididymal adipocyte surface area. Data are presented as mean and standard error. Normal: normal diet + phosphate-buffered saline (PBS); Control: 60% high-fat diet (HFD) + PBS; PPGK1: HFD + *P. pentosaceus* SMFM2017-GK1; LFNK3: HFD + *L. fermentum* SMFM2017-NK3; LPNK2: HFD + *L. plantarum* SMFM2017-NK2; LFNK4: HFD + *L. fermentum* SMFM2017-NK4. * Statistically significant, compared to the control group by pairwise *t*-test (*p* < 0.05).

**Figure 5 foods-10-00772-f005:**
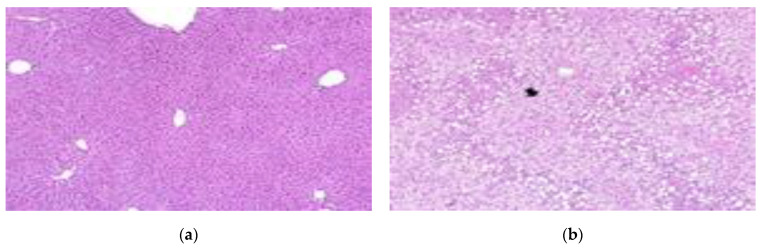
Representative microscopic images on liver tissues from mice groups (×100). (**a**) Normal: normal diet + phosphate-buffered saline (PBS), (**b**) Control: 60% high-fat diet (HFD) + PBS, (**c**) PPGK1: HFD + *P. pentosaceus* SMFM2017-GK1, (**d**) LFNK3: HFD + *L. fermentum* SMFM2017-NK3, (**e**) LPNK2: HFD + *L. plantarum* SMFM2017-NK2, (**f**) LFNK4: HFD + *L. fermentum* SMFM2017-NK4.

**Figure 6 foods-10-00772-f006:**
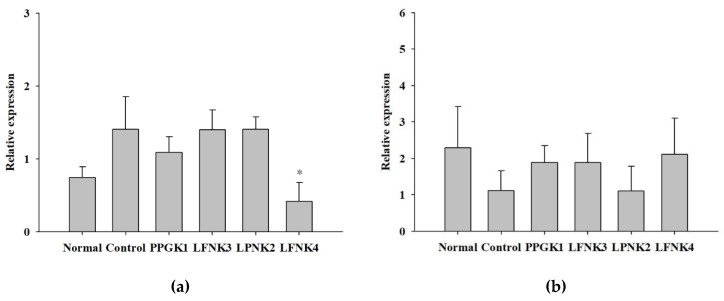
Relative expression levels of *Fas* (**a**) and *Cpt-2* (**b**) in liver tissues. Data are presented as mean and standard error. Normal: normal diet + phosphate-buffered saline (PBS); control: 60% high-fat diet (HFD) + PBS; PPGK1: HFD + *P. pentosaceus* SMFM2017-GK1; LFNK3: HFD + *L. fermentum* SMFM2017-NK3; LPNK2: HFD + *L. plantarum* SMFM2017-NK2; LFNK4: HFD + *L. fermentum* SMFM2017-NK4. * Statistically significant, compared to the control group by pairwise *t*-test (*p* < 0.05).

**Figure 7 foods-10-00772-f007:**
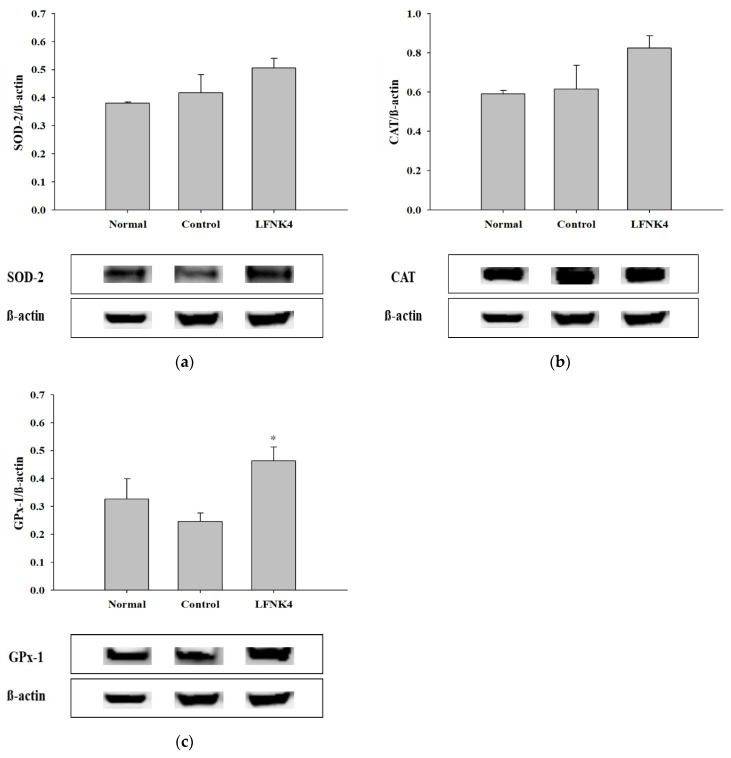
The immunoreactive intensity of SOD-2 (**a**), CAT (**b**), and GPx-1 (**c**) in liver tissues. Data are presented as mean and standard error. Normal: normal diet + phosphate-buffered saline (PBS); Control: 60% high-fat diet (HFD) + PBS; LFNK4: HFD + *L. fermentum* SMFM2017-NK4. * Statistically significant, compared to the control group by pairwise *t*-test (*p* < 0.05).

**Figure 8 foods-10-00772-f008:**
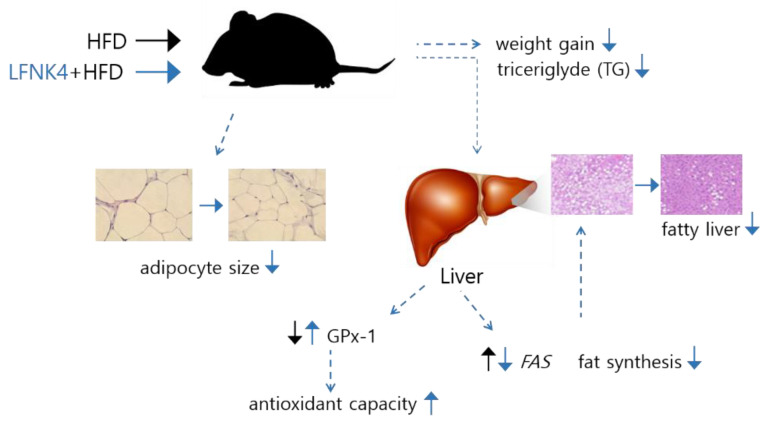
Effect of *L. fermentum* SMFM2017-NK4 administration on the improvement of fatty liver and lipid accumulation. A high-fat diet (HFD) causes fatty liver by increasing the expression of *Fas* gene in the liver. In addition, HFD decreases antioxidant capacity by decreasing the expression of GPx-1 protein in the liver. After administration of *L. fermentum* SMFM2017-NK4, fat synthesis was inhibited by decreasing *Fas* expression in the liver. In addition, antioxidant capacity was improved by increasing GPx-1 expression in the liver. *L. fermentum* SMFM2017-NK4 eventually inhibited the development of fatty liver and lipid accumulation.

**Table 1 foods-10-00772-t001:** Strains of 31 of lactic acid bacteria used for in vitro test.

Lactic Acid Bacteria Strains	Identification
1, 44	*Lactobacillus pentosus*
3, 28, 29, 31, 32, 66, 70, 73, 52-1-2	*Lactobacillus fermentum*
6, 50, 76, 77, 78, 79, 38-Y	*Lactobacillus plantarum*
7, 9, 11, 20	*Pediococcus pentosaceus*
12, 75	*Weissella cibaria*
26, 30	*Lactobacillus sakei*
45, 72	*Lactobacillus* spp.
71, 4Y	*Lactobacillus brevis*
98-1-3	*Lactobacillus curvatus*

**Table 2 foods-10-00772-t002:** Primer sequences of obesity related to genes used for qRT-PCR.

Gene	Primer Sequence (5′→ 3′)		Reference
*β-actin*	Forward	CCG TGA AAA GAT GAC CCA GAT C	Kim et al., 2012 [20]
	Reverse	CAC AGC CTG GAT GGC TAC GT	
*Fas*	Forward	TCT GGG CCA ACC TCA TTG GT	Ritze et al., 2014 [21]
	Reverse	GAA GCT GGG GGT CCA TTG TG	
*Cpt-2*	Forward	GCC TGC TGT TGC GTG ACT G	Yoo et al., 2013 [22]
	Reverse	TGG GTA CGA TGC TGT GC	

**Table 3 foods-10-00772-t003:** Inhibition ability of lactic acid bacteria for α-glucosidase activity.

Lactic Acid Bacteria Strains	Inhibition Rate	Lactic Acid Bacteria Strains	Inhibition Rate
66	41.45 ± 6.02 ^a^	72	29.02 ± 2.22 ^ab^
29	40.64 ± 3.76 ^ab^	32	27.85 ± 7.93 ^b^
50	39.13 ± 6.81 ^ab^	1	27.60 ± 16.90 ^b^
70	39.09 ± 4.49 ^ab^	7	27.35 ± 4.65 ^b^
75	37.22 ± 4.43 ^ab^	79	25.78 ± 9.50 ^b^
45	37.04 ± 4.04 ^ab^	52-1-2	25.69 ± 13.28 ^b^
44	34.64 ± 1.15 ^ab^	78	23.96 ± 9.31 ^bc^
73	33.11 ± 3.93 ^ab^	77	23.35 ± 4.69 ^bc^
20	30.68 ± 2.00 ^ab^	26	22.66 ± 2.96 ^bc^
98-1-3	30.16 ± 3.73 ^ab^	9	19.48 ± 5.29 ^bc^
31	30.00 ± 9.45 ^ab^	12	17.17 ± 9.19 ^bc^
38-Y	29.90 ± 10.90 ^ab^	3	15.23 ± 9.35 ^bc^
30	29.59 ± 9.94 ^ab^	11	12.27 ± 9.00 ^c^
76	29.52 ± 5.65 ^ab^	6	10.31 ± 12.71 ^c^
4Y	29.48 ± 10.71 ^ab^	28	6.58 ± 5.50 ^c^
71	29.02 ± 9.22 ^ab^		

Values are expressed as inhibition rate (%); mean ± standard deviation. ^a–c^ Different letters in a column indicate a significant difference (*p* < 0.05).

**Table 4 foods-10-00772-t004:** Inhibition ability of lactic acid bacteria for pancreatic lipase activity.

Lactic Acid Bacteria Strains	Inhibition Rate	Lactic Acid Bacteria Strains	Inhibition Rate
20	18.92 ± 4.27 ^a^	70	4.61 ± 0.94 ^bc^
11	18.80 ± 7.36 ^a^	44	3.25 ± 4.90 ^bc^
26	18.10 ± 1.76 ^ab^	12	2.91 ± 3.52 ^bc^
30	15.22 ± 6.93 ^ab^	79	2.76 ± 1.55 ^bc^
29	14.43 ± 5.27 ^ab^	45	2.40 ± 0.84 ^bc^
73	13.74 ± 4.44 ^ab^	71	2.24 ± 0.93 ^bc^
72	10.96 ± 5.39 ^ab^	66	2.13 ± 1.38 ^bc^
50	10.95 ± 5.87 ^ab^	78	2.05 ± 1.54 ^bc^
38-Y	10.39 ± 7.99 ^ab^	76	1.71 ± 2.83 ^bc^
31	10.15 ± 3.44 ^ab^	6	0.01 ± 6.87 ^bc^
4Y	9.03 ± 5.98 ^b^	9	−1.60 ± 3.89 ^c^
98-1-3	8.86 ± 8.91 ^bc^	75	−1.63 ± 4.30 ^c^
32	7.22 ± 3.74 ^bc^	52-1-2	−2.63 ± 3.54 ^c^
7	7.05 ± 0.84 ^bc^	3	−7.78 ± 7.01 ^c^
77	6.91 ± 6.84 ^bc^	28	−10.58 ± 16.38 ^c^
1	5.21 ± 4.52 ^bc^		

Values are expressed as inhibition rate (%); mean ± standard deviation. ^a–c^ Different letters in a column indicate a significant difference (*p* < 0.05).

**Table 5 foods-10-00772-t005:** Inhibition ability of lactic acid bacteria (LAB) strains (*L. fermentum* SMFM2017-NK4, *L. fermentum* SMFM2017-NK3, *P. pentosaceus* SMFM2017-GK1, and *L. plantarum* SMFM2017-NK2) for 3T3-L1 pre-adipocyte cell differentiation.

Lactic Acid Bacteria Strains		Inhibition Rate
73	*L. fermentum* SMFM2017-NK4	21.44 ± 6.70 ^a^
29	*L. fermentum* SMFM2017-NK3	20.99 ± 10.06 ^a^
20	*P. pentosaceus* SMFM2017-GK1	20.45 ± 2.53 ^a^
50	*L. plantarum* SMFM2017-NK2	15.57 ± 7.37 ^a^

Values are expressed as inhibition rate (%); mean ± standard deviation. ^a^ Letters in a column indicate no significant difference (*p* > 0.05).

**Table 6 foods-10-00772-t006:** Histopathological scores in liver tissue from mice groups.

	Groups					
	Normal	Control	PPGK1	LFNK3	LPNK2	LFNK4
Steatosis	±	+++	+++	++	+++	++

Normal: normal diet + phosphate-buffered saline (PBS); Control: 60% high-fat diet (HFD) + PBS; PPGK1: HFD + *P. pentosaceus* SMFM2017-GK1; LFNK3: HFD + *L. fermentum* SMFM2017-NK3; LPNK2: HFD + *L. plantarum* SMFM2017-NK2; LFNK4: HFD + *L. fermentum* SMFM2017-NK4. Grade; ±: minimal, ++: moderate, +++: marked.

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
