# Peer review of "Lactobacillus fermentum SMFM2017-NK4 Isolated from Kimchi Can Prevent Obesity by Inhibiting Fat Accumulation"

_foods, 2021, doi:10.3390/foods10040772_

Round 1

Reviewer 1 Report

Everything was corrected by the authors.

Author Response

Thank you for your affirmation of the quality of the manuscript.

Reviewer 2 Report

The article has been corrected according to Reviewer's comments and suggestions.

Author Response

Thank you for your reading and affirmation of the quality of the manuscript.

Reviewer 3 Report

The manuscript has been improved. Anyway, there are two minor points in the text that should be corrected:

L438-440 :"In our study, the expression levels of Cpt-2 were the highest in the 438 LFNK4 group among the lactic acid bacteria-treated groups, compared to the control 439 group although there is no statistically significant difference."

L460-461: "The expression levels of SOD-2 and CAT were higher in the LFNK4 group than 460 in the control group although there was no statistically significant difference .."

In both sentences, you should replace "were" by "looked, appeared, ..." to be more consistent with the non significance of the data.

Author Response

RESPONSES TO REVIEWER’S COMMENTS

The authors thank the editors and reviewers for constructive comments. Our responses to each of the comments are provided below.

Reviewer(s)' Comments to Author:

Reviewer 3

The manuscript has been improved. Anyway, there are two minor points in the text that should be corrected:

L438-440 :"In our study, the expression levels of Cpt-2 were the highest in the LFNK4 group among the lactic acid bacteria-treated groups, compared to the control group although there is no statistically significant difference."

L460-461: "The expression levels of SOD-2 and CAT were higher in the LFNK4 group than in the control group although there was no statistically significant difference .."

In both sentences, you should replace "were" by "looked, appeared, ..." to be more consistent with the non significance of the data.

Response: The authors corrected the words in lines 432 and 452.

This manuscript is a resubmission of an earlier submission. The following is a list of the peer review reports and author responses from that submission.

Round 1

Reviewer 1 Report

I am very impressed with the article submitted for review. Everything fits in it. The introduction provides all the necessary information, the results are described in a clear and clear way. The methodology was written very well, leaving no doubt as to how the determinations were performed. I have one comment regarding the sentence: "Therefore, L. fermentum SMFM2017-NK4 can be used as a probiotic for obesity."

You have carried out in vitro studies and on mice, is this really a sufficient scale to recognize that the bacteria can be used as a probiotic in human nutrition against obesity?

Author Response

Reviewer 1

Comments and Suggestions for Authors

I am very impressed with the article submitted for review. Everything fits in it. The introduction provides all the necessary information, the results are described in a clear and clear way. The methodology was written very well, leaving no doubt as to how the determinations were performed. I have one comment regarding the sentence: "Therefore, L. fermentum SMFM2017-NK4 can be used as a probiotic for obesity."

You have carried out in vitro studies and on mice, is this really a sufficient scale to recognize that the bacteria can be used as a probiotic in human nutrition against obesity?

Response: The authors agree the reviewer’s comments. For probiotic use, additional evaluation of probiotic candidates needs, for example, evaluation for safety of bacteria (antibiotics resistance and enzyme activity such as beta-glucuronidase activity) and experiment for human. Thus, the authors revised the sentences in lines 30 and 495-496.

Reviewer 2 Report

Manuscript ID : foods-1118175

Title : Lactobacillus fermentum SMFM2017-NK4 Isolated from Kimchi Can Prevent Obesity by Inhibiting Fat Accumulation.

Reading the article “Lactobacillus fermentum SMFM2017-NK4 Isolated from Kimchi Can Prevent Obesity by Inhibiting Fat Accumulation» was of great interest. I consider that the paper subject is in accordance with the scope of the journal Foods. The article is of interest regarding the importance of identifying new probiotic strains with health applications and particularly strains with anti-obesity capacities. Nevertherless, the article needs a hard work of proofreading.

In the introduction, a paragraph concerning inflammation is missing, regarding the experiment done and the data obtained.

In the materials and methods, please add a sentence explaining how you made the choice of the 4 strains to be studied (paragraph 2.5).

Generally, through the manuscript, your conclusions do not reflect your data. Figures 6 and 7, the statistical analysis do not support the text, you must be more cautious in your analysis. Please improve your text and point the difference between statistically significant data and the no significant.

Same remark for the figure 3. There is no evidence of different adipocyte cells sizes in the figure. You need to better explain what you can see and how you measure cells area.

Please extend the discussions in the whole paper. There is even no discussion point in the paragraph 3.1.

L248 : “that form adipocytes”, this sentence is confusing, please improve it.

Abbreviations like ALT, AST should be written in full.

The conclusion is too short and incomplete. Your data are over interpreted. Please improve the conclusion to make matches with your data.

Author Response

Reviewer 2

 Reading the article “Lactobacillus fermentum SMFM2017-NK4 Isolated from Kimchi Can Prevent Obesity by Inhibiting Fat Accumulation” was of great interest. I consider that the paper subject is in accordance with the scope of the journal Foods. The article is of interest regarding the importance of identifying new probiotic strains with health applications and particularly strains with anti-obesity capacities. Nevertherless, the article needs a hard work of proofreading.

  1. In the introduction, a paragraph concerning inflammation is missing, regarding the experiment done and the data obtained.

Response: The authors added a paragraph concerning inflammation in lines 53-57, as the reviewer’s comments.

  1. In the materials and methods, please add a sentence explaining how you made the choice of the 4 strains to be studied (paragraph 2.5).

Response: The authors added a sentence in lines 148-152 for explaining how the 4 strains were selected.

  1. Generally, through the manuscript, your conclusions do not reflect your data. Figures 6 and 7, the statistical analysis do not support the text, you must be more cautious in your analysis. Please improve your text and point the difference between statistically significant data and the no significant.

Response: Thank you for your constructive comments. The authors checked the statistical results again, and revised the sentence including statistically significant data (lines 439-442, 460, 467, 474-480, 491-496, and Figure 8).

  1. Same remark for the figure 3. There is no evidence of different adipocyte cells sizes in the figure. You need to better explain what you can see and how you measure cells area.

Response: The authors added the sentences to describe the methods and results for size of adipocyte cells (lines 187-188, 365-366, and 381-382), and revised the Figure 3e.

  1. Please extend the discussions in the whole paper. There is even no discussion point in the paragraph 3.1.

Response: The authore added the sentences for discussion in the paragraph 3.1 (lines 243-249). Also the authors added the sentences for discussion in the whole paper (lines 281-285, 360-363, 396-400, 420-422, and 424-432).

  1. L248 : “that form adipocytes”, this sentence is confusing, please improve it.

Response: The authors revised the sentence to clarify the meaning in lines 280-281.

  1. Abbreviations like ALT, AST should be written in full.

Response: The authors added the full name of ALT and AST in the Abstract and Introduction sections in lines 25 and 54.

  1. The conclusion is too short and incomplete. Your data are over interpreted. Please improve the conclusion to make matches with your data.

Response: The authors added and revised the conclusion with only statistically significant results not to over interpreted (lines 491-496).

Reviewer 3 Report

The article titled Lactobacillus fermentum SMFM2017-NK4 Isolated from Kimchi Can Prevent Obesity by Inhibiting Fat Accumulation presents anti-obesity effects of thirty-one lactic acid bacteria which were examined in vitro for their ability to inhibit α-glucosidase activity, lipase activity, and 3T3-L1 cell differentiation. From the studied group, four selected lactic acid bacteria were administered to obese C57BL/6J mice models for 8 weeks.

In my opinion the article could be very interesting for Foods readers. Introduction is well written and presents most important aspects that are helpful in understanding the aim of the researches.In the sentence (line 26-27) - Obesity is present in more than.....- the style is incorrect- please rearrange the sentence.

Section Material and methods is clearly presented, all the methods are presented in detail. As the researches deal with animal experiments, please inform, if You have a permission of bioethical commission. Please, inform, if approval by the Institutional Animal Care and Use Committee of Sookmyung Women's University is enough for such experiments in Your country.

Below I present some comments for Results/Discussion section.

  1. Table 1 is too long - it should be rearranged, for example: the same lactic bacteria strains could be placed in the same row of the table.
  2. The explanation of the symbols should be placed under the equation.
  3. Tables 3 and 4 are too long - they should be rearranged.
  4. Statistical analysis is not sufficient. Please complete the statistics with homogeneous groups (table 3, table 4, table 5).
  5. line 271 - it should be - weight gain, which is the difference...
  6. lines 276-279 - the style of the sentence is incorrect, the sentence is incomprehensible, please correct the sentence.
  7. lines 304-319 - the fragment of the text is hard to be read, because of the style of the sentences - all the sentences are very similar. The fragment should be rearranged.

In the article, Authors presented many interesting results, so they should be also considered in Conclusion section. In my opinion, this section should be completed. Figure 8 should be rather presented in Results/Discussion section.

Author Response

Reviewer 3

The article titled Lactobacillus fermentum SMFM2017-NK4 Isolated from Kimchi Can Prevent Obesity by Inhibiting Fat Accumulation presents anti-obesity effects of thirty-one lactic acid bacteria which were examined in vitro for their ability to inhibit α-glucosidase activity, lipase activity, and 3T3-L1 cell differentiation. From the studied group, four selected lactic acid bacteria were administered to obese C57BL/6J mice models for 8 weeks.

  1. In my opinion the article could be very interesting for Foods readers. Introduction is well written and presents most important aspects that are helpful in understanding the aim of the researches. In the sentence (line 26-27) - Obesity is present in more than.....- the style is incorrect- please rearrange the sentence.

Response: The authors revised the sentence in lines 38-39.

  1. Section Material and methods is clearly presented, all the methods are presented in detail. As the researches deal with animal experiments, please inform, if You have a permission of bioethical commission. Please, inform, if approval by the Institutional Animal Care and Use Committee of Sookmyung Women's University is enough for such experiments in Your country.

Response: In S. Korea, Institutional Animal Care and Use Committee of Sookmyung Women's University was established for purpose of protecting and ensuring the ethical handling of laboratory animals used in animal experiments within the university. It conforms and deliberates ANIMAL PROTECTION ACT and LABORATORY ANIMAL ACT of S. Korea. Therefore, the authors performed animal experiments after approval by IACUC of Sookmyung Women’s University.

  1. Below I present some comments for Results/Discussion section.

Table 1 is too long - it should be rearranged, for example: the same lactic bacteria strains could be placed in the same row of the table. The explanation of the symbols should be placed under the equation. Tables 3 and 4 are too long - they should be rearranged.

Response: The authors rearranged Tables 1, 3 and 4, and the explanation of the symbols was replaced under the equation in lines 143-145.

  1. Statistical analysis is not sufficient. Please complete the statistics with homogeneous groups (table 3, table 4, table 5).

Response: The authors added the statistical results in Table 3, 4, and 5.

  1. line 271 - it should be - weight gain, which is the difference...

Response: The authors revised the sentence in line 304.

  1. lines 276-279 - the style of the sentence is incorrect, the sentence is incomprehensible, please correct the sentence.

Response: The authors corrected the sentences in lines  309-313.

  1. lines 304-319 - the fragment of the text is hard to be read, because of the style of the sentences - all the sentences are very similar. The fragment should be rearranged.

Response: The authors rearranged and revised the several sentences in lines 335-348.

  1. In the article, Authors presented many interesting results, so they should be also considered in Conclusion section. In my opinion, this section should be completed. Figure 8 should be rather presented in Results/Discussion section.

Response: Thank you for the reviewer’s constructive comments. The authors added all the significant results of this study in the Conclusion section (lines 491-496), and Figure 8 and the related sentences (lines 474-480) were moved to Results/Discussion section.